# The Challenging Diagnosis of Pediatric Multisystem Inflammatory Syndrome Associated with Sars-Cov-2 Infection-Two Case Reports and Literature Review

**DOI:** 10.3390/jpm11040318

**Published:** 2021-04-19

**Authors:** Marcela Daniela Ionescu, Roxana Taras, Bianca Dombici, Mihaela Balgradean, Elena Camelia Berghea, Alin Nicolescu

**Affiliations:** 1Department of Pediatrics, Carol Davila University of Medicine and Pharmacy, 020021 Bucharest, Romania; daniela.ionescu@umfcd.ro (M.D.I.); roxana.taras@drd.umfcd.ro (R.T.); bianca.dombici@rez.umfcd.ro (B.D.); mihaela.balgradean@umfcd.ro (M.B.); 2Marie Curie Emergency Children’s Hospital, 041451 Bucharest, Romania; nicolescu_a@yahoo.com

**Keywords:** multisystem inflammatory syndrome in children, cytokines, Kawasaki disease, pediatric COVID19—Kawasaki

## Abstract

Severe acute respiratory coronavirus 2 (SARS-CoV-2) is a novel coronavirus discovered in 2019 that caused the coronavirus disease 2019 (COVID19). During the last year, over 70 million people were infected and more than 1.5 million people died. Despite the tremendous number of people infected, children were less affected and presented milder forms of the disease. A short time after the pandemic was declared, a new hyperinflammatory syndrome resembling Kawasaki disease (KD) was described in children with confirmed or suspected SARS-CoV-2 infection named multisystem inflammatory syndrome in children (MIS-C). The incidence of MIS-C is low and it has a polymorphous clinical presentation, making the diagnosis difficult. Although the incidence is reduced, there is a high risk of cardiovascular complications. In order to raise awareness, we present the cases of two pediatric patients diagnosed with MIS-C in our clinic.

## 1. Introduction

During the last year, the entire world confronted the pandemic caused by the SARS-CoV-2 infection, first reported in Wuhan, China, which rapidly spread worldwide and caused catastrophic damages [1,2]. Although until this moment, over 70 million people have been infected, and more than 1.5 million people have died, children have been less affected [3]. In children, the SARS-CoV-2 infection produces mild, primarily respiratory or gastrointestinal symptoms in contrast to the severe forms in adult patients [4,5]. Despite the mild forms of the disease reported within pediatric population, a rare hyperinflammatory syndrome associated with coronavirus infection was described, initially named pediatric multisystem inflammatory syndrome temporally associated with SARS-CoV-2 infection (PIMS-TS) in Europe and multisystem inflammatory syndrome in children (MIS-C) in the United States [4,6,7]. The MIS-C clinical and paraclinical presentation has common characteristics with the presentation of KD, including fever, high levels of inflammatory markers, and multisystem damage (e.g., hematologic, dermatologic, cardiac) [8,9]. Large case series of KD related to SARS-CoV-2 infection from the United Kingdom (UK), Italy, the United States (USA), and France were published last year [1,4,9,10,11,12]. The MIS-C diagnosis can be difficult due to its polymorphous presentation and the shortage of knowledge of a previously SARS-CoV-2 infection which may be asymptomatic in pediatric patients [13,14]. Compared to KD, MIS-C has a higher risk of cardiovascular derangement and a higher rate of mortality therefore, it needs prompt diagnosis and treatment [4]. Of cardiovascular complications, specific data about arrhythmias are yet to be published [15]. From the information published so far, it is worth remembering that atrial flutter is very rarely described, both in adult and pediatric patients with COVID19, and the onset of this type of arrhythmia represents a negative biomarker predicting mortality [16,17].

To increase the degree of recognition of the potential for severe evolution and diagnostic framing difficulties, we selected 2 MIS-C cases from the Department of Pediatrics of Marie Curie Children’s Emergency Hospital, Bucharest, Romania.

### 1.1. Case Presentation #1

We present the case of a 4 year and 9-month-old male patient, who was admitted to the pediatric department of “Marie Curie” Emergency Children’s Hospital, for fever (39.2 °C) which started 2 days before hospitalization, associated with generalized macular erythematous skin rash.

The history of present illness does not include any upper or lower respiratory tract infection or any gastrointestinal infection in the last 3 months nor a close contact with confirmed or suspected COVID 19 individuals in the last 4 weeks.

In the emergency department, the vital signs included a temperature of 38 degrees, respiratory rate of 20 breaths/min, oxygen saturation of 99% on room air, heart rate of 135 beats/min, and blood pressure of 95/51 mmHg. In the emergency department, the vital signs included a temperature of 38 °C, respiratory rate of 20 breaths/min, oxygen saturation of 99% on room air, heart rate of 135 beats/min, and blood pressure of 95/51 mmHg.

The physical examination revealed a sick-appearing patient, with periorbital edema and non-exudative conjunctivitis, dry cracked lips, (See Figure 1), generalized non-pruritic macular erythematous skin rash, and erythematous edema affecting the extremities (See Figure 2). The pharynx was hyperemic, without any notably palpable lymphadenopathy. He had clear lungs to auscultation. The patient was tachycardic, without any murmurs to auscultation. The extremities were warm and the capillary refill time was of 2s. The abdominal palpation did not reveal any tenderness.

Of the imaging analysis, the chest X-ray was unremarkable. The abdominal ultrasonography revealed a few mesenteric lymph nodes with a maximum diameter of 23/14 mm and 5 mm of free fluid in the recto-vesical pouch. The echocardiography did not reveal any coronary modifications and the electrocardiogram was normal.

Due to the clinical presentation (fever, conjunctivitis, macular erythematous rash, erythematous burning, and swollen lips, peripheral edema) and the results of laboratory tests, the diagnosis of vasculitis was suspected, including KD. In the absence of prolonged fever, absence of lymphadenopathies, and echocardiography without any coronary abnormalities, KD was excluded.

The significant inflammatory syndrome (C-reactive protein-CRP-186.11 mg/dL, procalcitonin-413.9 ng/mL, ferritin-218 ng/mL), the high level of interleukin-6 (IL-6=31.51 pg/mL), and the elevated D-dimer levels (1.96µg/mL), raised the suspicion of a multisystem inflammatory syndrome post-COVID19 infection.(See Table 1) The immunochromatographic detection of SARS-CoV-2 serum antibodies was positive for immunoglobulin G (IgG) and negative for immunoglobulin M (IgM), suggesting a previous SARS-CoV-2 infection.

This clinical pattern represents a new phenomenon affecting previously asymptomatic children with SARS-CoV-2 infection manifesting as a hyperinflammatory syndrome with multiorgan involvement similar to Kawasaki disease [15].

The diagnosis of a pediatric multisystem inflammatory syndrome associated with SARS-CoV-2 infection was established, the patient fulfilling the Center for Disease Control (CDC) definition criteria (See Table 2).

Since clinical presentation and the laboratory findings resembled KD, we decided to start the management of the case based on KD treatment principles. The patient received intravenous fluid resuscitation for 5 days, intravenous immunoglobulin (400 mg/day for 5 days), corticosteroids (2 mg/kg/day, with decreasing dose, for 13 days) and empirical broad-spectrum antibiotics (Meropenem for 12 days and Targocid for 14 days) according to the MIS-C treatment guideline [16,17,18]. Symptomatology and laboratory tests slowly improved during hospitalization, with the remission of fever, rash, peripheral edema, conjunctivitis, normalization of the coagulogram, and the decrease of inflammatory markers values. The outcome was favorable, the patient did not have any cardiovascular complications even if he was at high risk.

### 1.2. Case Presentation #2

We present the case of a 15-year-old male patient transferred to our clinic for fever, vomiting, and chest pain.

The history of presenting complaint includes fever (40 °C), vomiting and somnolence, started 6 days before hospitalization. The patient received at-home treatment prescribed by the general practitioner consisting of antibiotics, antipyretics, and antiemetics. The clinical state worsened, the patient alleged chest pain and palpitations, so he presented to the territorial hospital. Laboratory tests were performed revealing leukopenia, lymphopenia, thrombocytopenia, an important inflammatory syndrome (CRP = 235 mg/L), and an elevated brain natriuretic peptide value (BNP) of 344 pg/mL. The electrocardiogram (ECG) was suggestive for atrial flutter (See Figure 3). The chest radiography showed aspects of peribronchovascular interstitial lung edema (See Figure 4). The patient was transferred in our clinic.

The initial vital signs included fever (38.7 °C degrees), tachypnea (respiratory rate of 35 breaths/min), hypoxemia (oxygen saturation of 90–93% on room air), tachycardia (heart rate of 130 beats/min), and normal blood pressure (100/67 mmHg).

The clinical examination revealed a sick appearing patient, conscious, afebrile at the moment, without any rash or lymphadenopathies. He had diminished vesicular murmur to the bases, tachypnea, but no rales. He was tachycardic (130–150 bpm), without any cardiac murmur and he complained of chest pain. The time of capillary refill was <2 s. He had nausea without vomiting and he alleged diffuse pain to abdominal palpation.

The laboratory tests show normal white blood cell count, lymphopenia, thrombocytopenia and increased values of inflammatory markers (CRP-274.53 mg/L, erythrocyte sedimentation rate-ESR-47mm/L, procalcitonin-2.04 ng/mL, IL-6-113.9 pg/mL, ferritin-3331 ng/mL, fibrinogen 555 mg/dL) (See Table 3).

The NT-proBNP was elevated (6421 pg/mL) and the troponin T, creatine kinase (CK) and CK-MB were normal. The D-dimers were positive (2.38 µg/mL), and the coagulogram showed a prolonged partial thromboplastin time (PTT 43.2 s), suggesting consumptive coagulopathy. Increased transaminases and the gamma-glutamyl-transpeptidase (GGT) revealed liver dysfunction. The elevated levels of serum creatinine concentration (1.28 mg/dL), the serum urea (76.5 mg/dL), and proteinuria (0.68 g/24 h) suggested renal involvement. The ECG was suggestive for atrial flutter (atrial rate of 300 beats/min, ventricular rate of 100–150 bpm, 3:1–2:1 atrioventricular conduction). The echocardiography revealed mild depression of left and right ventricular function (left ventricular ejection fraction of 50% and right ventricular ejection fraction of 45%), small mitral and tricuspidian regurgitation, and a small pericardial effusion.

The abdominal echography was unremarkable, and the thoracic-abdominal computer tomography (CT) scan showed interstitial lung edema, distributed bilaterally on the bases, bilateral pleural effusion (10 mm on the left side, 12 mm on the right side), a small pericardial effusion and cardiomegaly, with a cardiothoracic ratio of 0.6, without any abnormalities of the liver or spleen (See Figure 5, Figure 6 and Figure 7).

The pericardial effusion, bilateral pleural effusion, along with the ventilation disorders described on the echocardiography and CT scan, are imaging findings demonstrating the congestive heart failure that appeared as a complication of the atrial flutter.

The atrial flutter is very unusual in adolescents, and this may be an indicator of a viral infection or sepsis associated with an exaggerated immune response that can cause myocardial inflammation and, consequently, arrhythmias [15,17,19]. Blood cultures and urine cultures were negative, as were serology for *Influenza Virus* and *Mycobacterium tuberculosis.*

From the family history, we found that both grandmother and mother had respiratory infection 1 month ago. Furthermore, the patient alleged anosmia and ageusia during the same period. Considering his history and the actual epidemiological context, he was tested for SARS-CoV-2 infection (polymerase chain reaction-PCR and IgM and IgG antibodies), with negative PCR test result and positive results for IgG antibodies. The MIS-C diagnosis was established using the CDC criteria (see Table 2).

Due to the clinical presentation with cardiac complications, this case was classified as severe and the treatment administered was accordingly to the MIS-C treatment guidelines [15,18].

The priority was the sinus rhythm restoration. A 50 Joules electrical shock was applied under sedation, followed by antiarrhythmic therapy (Amiodarone 400 mg/day for 5 days, then 200 mg/day for 14 days) with a significant improvement in cardiac performance. The treatment of the inflammation consisted in administration of corticotherapy (2 mg/kg/day for 7 days, and progressively decreasing doses for 14 days), intravenous immunoglobulins 20 g/day for 1 day, followed by recombinant IL-1-receptor antagonist (IL1RA) Anakinra 100 mg/day for 7 days. The patient received also antibiotherapy (Ceftriaxon 2 g/day) for 11 days and thromboprophylaxy (Enoxaparine 40 mg/day) for 14 days, with clinical and paraclinical improvement. He did not experience another episode of atrial flutter, the fever subsided, the oxygen saturation maintained in normal ranges, the values of inflammatory markers decreased, also the cardiac enzymes and the echocardiography showed improvement of the cardiac function.

## 2. Literature Review

### 2.1. Coronavirus Disease 2019

COVID-19 is an illness produced by SARS-CoV-2 infection, described to affect mostly adults and only a small proportion of children [18,19,20]. Almost 90% of children and adolescents were described to have an asymptomatic or mild form of the disease that does not require any medical intervention [4,5,19]. (p. 19). The mortality rate in pediatric patients is <1% [8,21]. Despite this favorable outcome, in April 2020, a group of clinicians in the UK reported the first cases of hyper inflammation, fever, and cardiovascular shock in 8 previously healthy children [2,8]. Of these patients, all had a significant inflammatory syndrome and negative tests for an acute SARS-CoV-2 infection, although many had recent COVID-19 contact [8]. The clinical presentation of these patients was related to other pediatric inflammatory syndromes such as Kawasaki disease and toxic shock syndrome [20,21,22]. Firstly, this hyperinflammatory syndrome was considered to be a Kawasaki variant [8].

### 2.2. Kawasaki Disease

Kawasaki disease is a self-limited childhood vasculitis affecting the small and medium-sized vessels, including the coronary arteries [4,23]. In the absence of treatment, almost a quarter of children develop coronary artery aneurysms [24]. The etiology remains unknown, but recent or active infections in genetically predisposed children have been incriminated [2,8]. KD is found worldwide, the incidence being higher in Asian individuals and affecting children less than 5 years old [4]. The clinical manifestations required for the diagnosis of classical KD include high fever, unresponsiveness to antipyretics lasting for more than 5 days, and at least 4 of 5 of the following criteria: bilateral non-exudative conjunctivitis, polymorphous generalized rash, cervical lymphadenopathy (<1.5 cm), peripheral extremity changes (erythema of palms, edema of hands and feet, peeling of fingers or toes) and oropharyngeal changes (strawberry tongue, erythematous cracking lips) [24]. Sometimes, besides fever lasting for more than 4 days, only 2 or 3 of the other clinical criteria can be met. The association of fever >4 days with 2 or 3 criteria diagnoses an incomplete form of KD. The most severe variant of KD is the KD shock syndrome associated with hypotension unresponsive to treatment [25]. In order to prevent coronary complications, intravenous immunoglobulin and aspirin treatment must be initiated promptly [8].

### 2.3. MIS-C

The CDC defines MIS-C as a syndrome that affects individuals aged <21 years positive for current or recent SARS-CoV-2 infection (PCR, serology, or antigen test) or considered close contacts to suspected or confirmed COVID19 cases within the 4 weeks prior to the onset of symptoms which develop fever ≥38.0 °C for more than one day and have laboratory evidence of inflammation (high levels of CRP, ESR, procalcitonin, fibrinogen, lactate dehydrogenase-LDH, IL-6, d-dimer, ferritin, neutrophilia, lymphopenia, hypoalbuminemia), and evidence of clinically severe illness requiring hospitalization, with multisystem (≥2) organ involvement (cardiac, renal, respiratory, hematologic, gastrointestinal, dermatologic or neurological); in the absence of other plausible diagnoses [16].

By comparison with the rate of SARS-CoV-2 infection in patients aged <21-year-old (322 in 100,000), MIS-C is rarely seen (2 in 100,000) [25,26]. However, the awareness of this new syndrome must be raised due to its polymorphous presentation and its possible lethal complications.

Initially, MIS-C presentation appears similar to KD or toxic shock syndrome, the symptomatology including fever, rash, conjunctivitis, and sometimes shock if the myocardium is involved [6]. Compared to KD, gastrointestinal symptoms are more common in MIS-C. In order to prevent complications like coronary aneurysms and myocardial damage, the diagnosis should be established quickly and the treatment administered immediately (See Figure 8).

Until this moment, there does not exist any standardized treatment protocol. The treatment prescribed in the reported case series was similar to the treatment of KD (intravenous immunoglobulin, low-dose aspirin) along with corticosteroids and broad-spectrum antibiotics. In patients with circulatory shock, vasoactive support and fluid resuscitation were required [1,2,4].

### 2.4. KD versus MIS-C

All the case series reported in London, then Bergamo, New York City, and Paris helped to define the MIS-C and to differentiate it from KD. The resemblance between KD and MIS-C is that they have various clinical presentations and none of them are associated with a pathognomonic diagnostic test [2,27]. The differences between the two entities consist in the race most commonly involved (the Asian race in KD and the African in Kawasaki-like syndrome), the males are more frequently affected in MIS-C, the age of onset (KD appears mostly in <5-year-old children, and MIS-C in children >4-year-old and adolescents), the symptomatology, MIS-C being associated mostly with an incomplete form of KD [4,28]. Furthermore, a more frequent gastrointestinal involvement and a higher risk of cardiovascular complications in MIS-C (76%) was found than in KD (26–40% in the absence of the treatment and <3% if immunoglobulins are administered) like coronary aneurysms, myocarditis with cardiogenic shock [9,15]. Additionally, the laboratory tests reveal different abnormalities (in MIS-C, the blood count reveals frequently leukopenia with neutrophilia and lymphopenia, and thrombocytopenia, in KD lymphopenia being rare) [4,8,29]. The level of inflammatory markers is much more elevated in MIS-C than in KD. Concerning the treatment, patients with MIS-C may present greater resistance to intravenous immunoglobulin [4].

### 2.5. COVID19 and Atrial Flutter

Most MIS-C cases associate cardiovascular complications, such as myocardial dysfunction, myocarditis, coronary dilation or aneurysm, atrio-ventricular block, or arrhythmias [30]. Arrhythmias are poorly described in children and may have nonspecific manifestation such as premature ventricular beats, premature atrial beats, changes of the ST segment, or QT prolongation [16,30]. Atrial fibrillation and atrial flutter are rarely reported [30,31]. In adult patients with COVID19, atrial fibrillation and atrial flutter incidence are also decreased (10%) [32]. Atrial flutter development in adolescents may be an indicator of a viral infection or sepsis associated with an exaggerated immune response who can cause myocardial inflammation and, as a consequence, arrhythmias [31] (p. 2). Although rare, the onset of atrial flutter is associated with high rate of mortality and poor outcomes, so it needs rapid recognition and intervention [17,32].

## 3. Discussion

Almost 20% of children infected with SARS-CoV-2 are asymptomatic [33]. Independent of the clinical syndrome, the MIS-C and non-MIS-C patients develop an immunological response consisting of high levels of IgG and immunoglobulin A (IgA) and absence or low levels of circulating IgM [34,35]. The number of IgG levels increased over time after the onset of symptoms and has a few weeks of stability, this durable response being demonstrated in MIS-C patients. The IgA levels exceed the levels found in adults’ convalescent plasma and may explain the gastrointestinal symptoms in children [34].

Because MIS-C appears weeks later after the SARS-CoV-2 infection, it is supposed that an aberrant immune response is involved [26]. The serum profiling reveals high levels of IL-1β, IL-6, IL-8, IL-10, IL-17 and gamma interferon (IFN-ϒ) and normal or low ranges of alpha tumor necrosis factor (TNF-α) [30,31]. MIS-C, compared with KD, have endothelial dysfunction associated with mild and transient coronary dysfunction, without any morphological changes as aneurysms. This type of coronary dilation is similar to the febrile onset of juvenile systemic arthritis [32,33,34,35].

In KD, the pathophysiological mechanism includes the production of autoantibodies as a response to an acute viral infection. This response involves mostly the IgA producing plasma cells. In the artery walls, IgA producing cells and neutrophils were found. By comparison with MIS-C, in KD these cells contribute to morphology changes in the arterial wall and destruction of the connective tissue with the development of arterial aneurysms [30,36,37]. The serum profiling reveals elevated values of IL-17A, IL-6, C-X-C motif chemokine ligand 10 (CXCL10) and higher values of markers associated with coronary artery disease [30,31].

The similarities and differences of the pathophysiological mechanisms might influence the treatment choices. Intravenous immunoglobulins may neutralize the autoantibodies produced in both diseases. Corticosteroids help to generate general immunosuppression. Due to the high levels of IL-17A, KD secukinumab can be considered. Instead, MIS-C can be treated with anakinra, a recombinant IL-1-receptor antagonist and also with IL-6 antagonists. Anti-TNF-α medicines are not effective due to the normal range of this biomarker [30].

### Case Particularities

The first patient presented an incomplete form of KD (fever <2 days, rash, conjunctivitis, peripheral edema and oropharyngeal modifications) and positive IgG antibodies anti-SARS-CoV-2, even though he had not had any symptoms or close contact with a COVID19 individual. He was one of the first patients hospitalized in our clinic for MIS-C. The symptomatology resembled the cases described initially in the literature, but comparatively the patient did not allege any gastrointestinal symptoms [38]. The limitations of the laboratory did not allow us to dose the levels of IgA, which could have explained the absence of gastrointestinal symptoms. The inflammatory markers were incredibly elevated, especially procalcitonin (413.9 ng/mL), compared to the mean value reported in the literature 30.5 ± 2.1 [39]. High levels of inflammatory markers are associated with Kawasaki disease shock syndrome [4]. Ahmed et al. report that 54% of the MIS-C patients associate cardiac complications with echocardiography modifications, and 50% develop cardiogenic shock. Moreover, 71% of patients need hospitalization in the ICU department [39]. Although our patient had a high risk of developing cardiac complications and shock, his evolution was favorable. 

The second patient described COVID19-specific symptoms 4 weeks before hospitalization and he was part of a community known to have a high rate of SARS-CoV-2 infection. Although the MIS-C have clinical manifestations of complete KD in 64% of cases, our patient presented only fever as a KD criteria [19]. Instead, he had more severe manifestations, such as atrial flutter. The atrial flutter is very unusual in adolescents and extremely rare in the literature in MIS-C patients [31,40]. Atrial flutter onset is associated with an elevated risk of mortality, so immediate recognition and treatment are required [32]. We chose to present this case in order to highlight that the symptomatology in adolescents with MIS-C can be more severe, with multiorgan dysfunction, as described in COVID19-related multisystem inflammatory syndrome in adults [41].

## 4. Conclusions

Multisystem inflammatory syndrome in children associated with COVID19 is a rare and severe complication of SARS-CoV-2 infection with polymorphous presentation. To highlight the variability of the clinical presentations, we exposed two cases of MIS-C, a case with Kawasaki-like symptoms and a case with atrial flutter and cardiac involvement. MIS-C must be diagnosed immediately, being a disease that requires aggressive treatment initiated as soon as possible to prevent possible complications (coronary aneurysms, myocarditis, and cardiogenic shock). Initially considered a KD variant, MIS-C may have a worse prognosis than KD so a high degree of vigilance must be maintained during this pandemic.

This report showed that inflammatory markers concentration (CRP, ESR, procalcitonin, ferritin, IL-6, fibrinogen), and D dimers value, can be used as laboratory findings for diagnosis, for the appropriate treatment choice, and for monitoring disease improvement in COVID-19 as well. We also encourage further studies to make a prognostic model that includes these biomarkers along with other proven poor prognostic factors in similar cases.

## Figures and Tables

**Figure 1 jpm-11-00318-f001:**
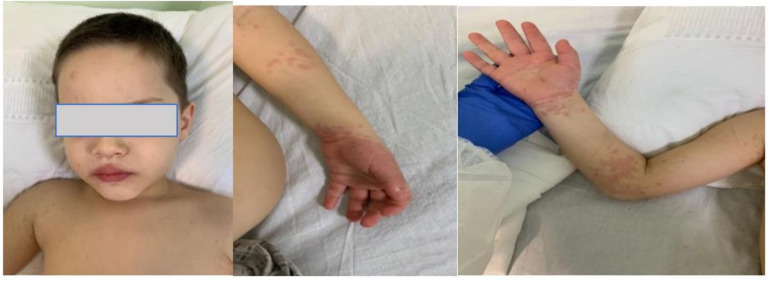
Periorbital edema. Cracked lips. Upper extremity edema and erythema.

**Figure 2 jpm-11-00318-f002:**
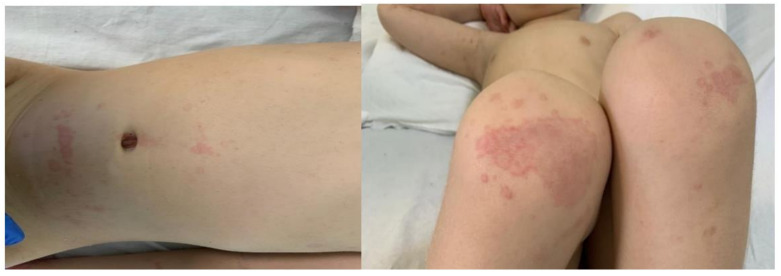
Maculo-erythematous skin rash. (All the images have parental permission).

**Figure 3 jpm-11-00318-f003:**
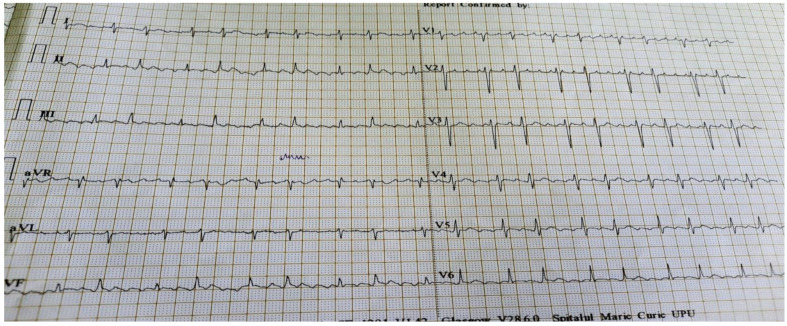
ECG suggestive for atrial flutter with an atrial rate of 300 beats per min (bpm). Atrioventricular conduction rate is variable at 2:1 and 3:1. Therefore, the ventricular rate ranges from 100–150 bpm.

**Figure 4 jpm-11-00318-f004:**
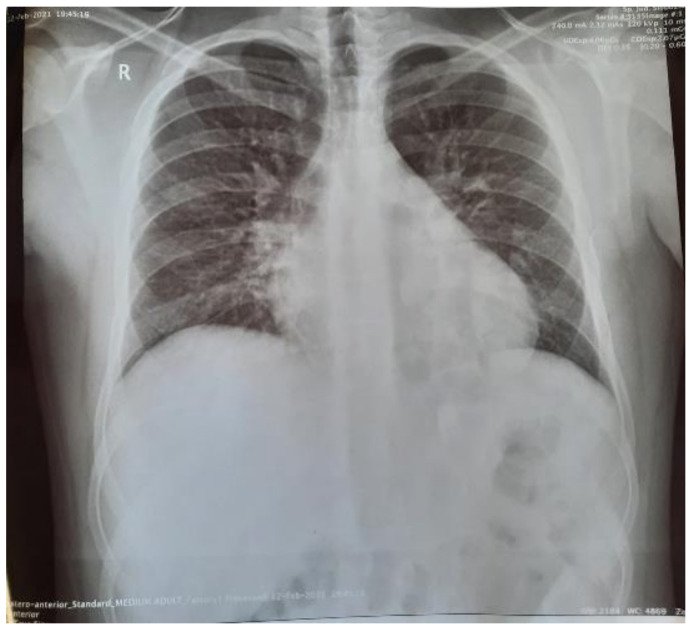
Chest radiography suggestive for aspects of peribronchovascular interstitial edema.

**Figure 5 jpm-11-00318-f005:**
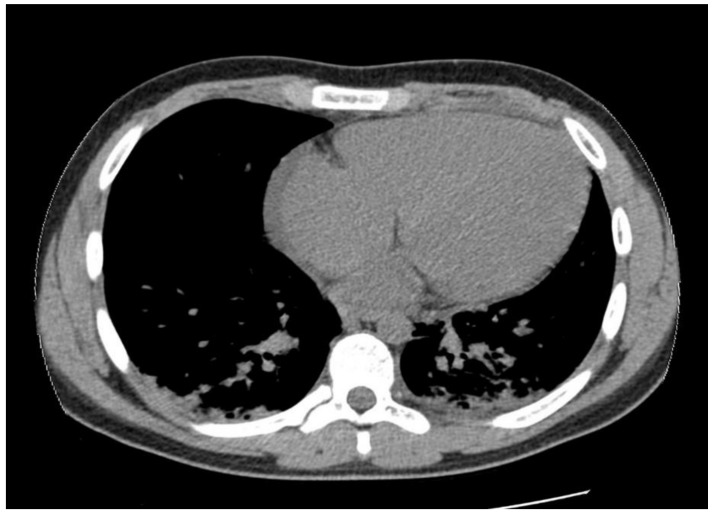
Increased cardiothoracic ratio. Pericardial effusion.

**Figure 6 jpm-11-00318-f006:**
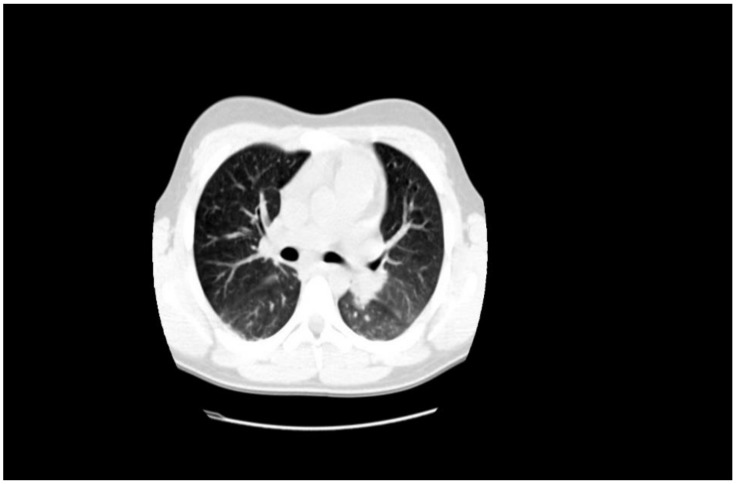
Interstitial edema in congestive heart failure.

**Figure 7 jpm-11-00318-f007:**
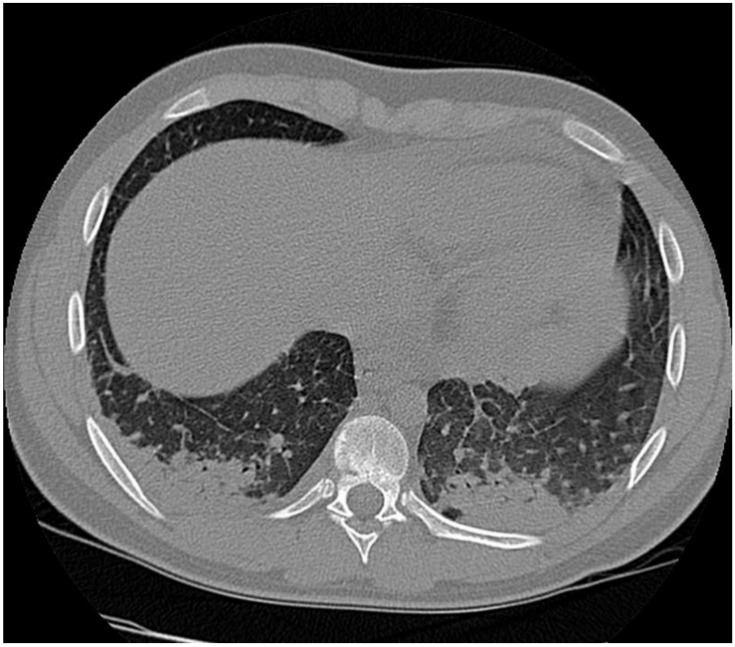
Interstitial edema in congestive heart failure. Small bilateral pleural effusion.

**Figure 8 jpm-11-00318-f008:**
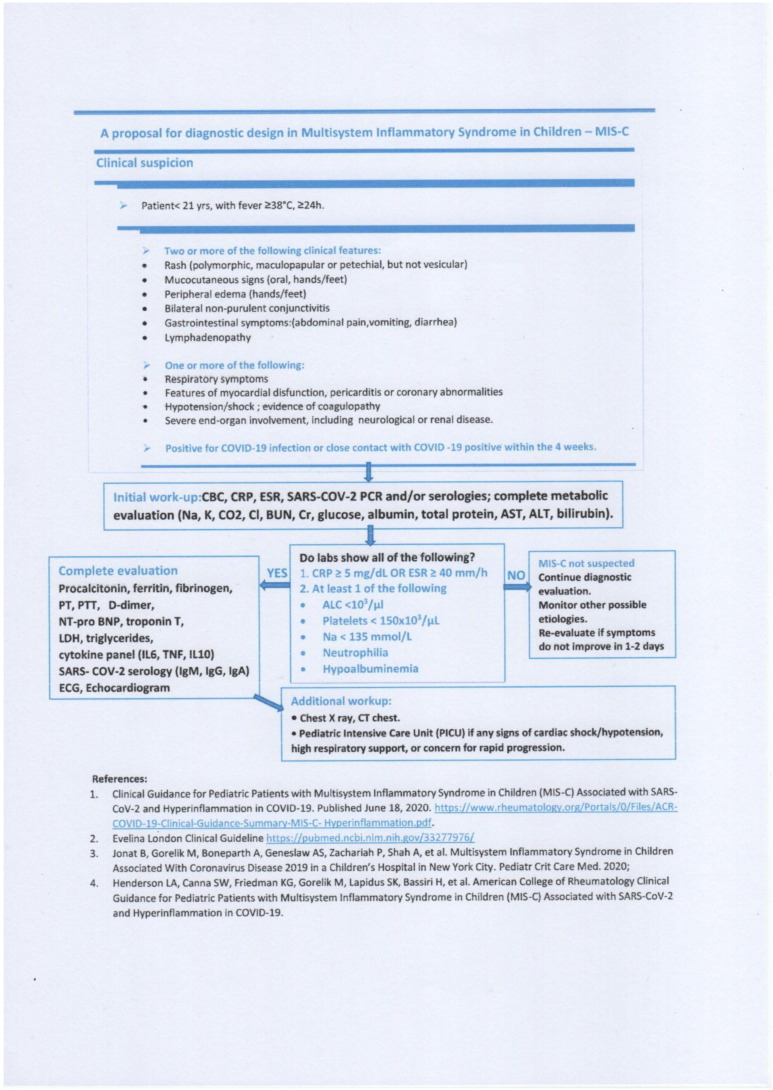
A proposal for diagnostic design in Multisystem Inflammatory Syndrome in Children.

**Table 1 jpm-11-00318-t001:** Case 1—Basal and after treatment laboratory parameters.

	Before Treatment	After Treatment	Normal Range
Complete blood count			
White blood cells count	12.12	10.62	5.50–15.50 × 10^3^/µL
Lymphocytes	1.2	2.14	2–8 × 10^3^/µL
Neutrophils	10.38	7.42	1.5–8.5 × 10^3^/µL
Thrombocytes	94000	554000	150.000–450.000/µL mm^3^
Hb	13.10	12.3	11–14 g/dL
Rheumatology			
C-reactive protein	186.11	0.31	0–5 mg/L
Procalcitonin	413.9	0.16	<0.05 ng/mL
Ferritin	218		4–67 ng/mL
IL-6	31.51		<7 pg/mL
LDH	251	223	120–300 U/L
Myocardium function			
PT	22.1	15.5	11.3–15.6 s
APPT	38.2	29.5	24–37 s
INR	1.7	1.17	0.84–1.2
D-dimers	1.96	0.49	0–0.5 µg/mL
Troponin T	<40	<40	<40 ng/mL
NT proBNP	200	187	<125 pg/mL
CK	32	15	7–149 U/L
CK-MB	23.5	22.5	7–25 U/L
Kidney function			
Creatinine	0.39	0.29	<0.47 mg/dL
BUN	30.7	29.2	<39 mg/dL
Ionogram			
Na^+^	129	136.3	138–145 mmol/L
K^+^	3.97	4.45	3.5–5.1 mmol/L
Liver function			
TGO	23.5	27.4	2–48 U/L
TGP	27.1	30.5	2–29 U/L
Albuminemia	3.35	3.75	3.8–5.4 g/dL
Proteinemia	5.27	6.2	6–8 g/dL
Infectious disease			
Blood cultures	Negative		Negative
Urine culture	Negative		Negative
PCR SARS-CoV-2	Negative		Negative
IgM SARS-CoV-2	Negative		Negative
IgG SARS-CoV-2	Positive		Negative

**Table 2 jpm-11-00318-t002:** CDC diagnostic criteria [16].

Age < 21 years Fever ≥ 24 h (>38 °C or subjective fever) Clinically severe disease necessitating hospitalization Laboratory tests suggesting inflammation (CRP, ESR, procalcitonin, fibrinogen, D-dimer, LDH, IL-6, ferritin, hypoalbuminemia, lymphopenia, neutrophilia) ≥ 2 systems involved (cardiac, respiratory, renal, neurological, dermatologic, hematologic, gastrointestinal) **AND**
No alternative plausible diagnoses **AND**
Positive for current or recent SARS-CoV-2 infection (RT-PCR, antigen test, serology **or** Exposure to a suspected or confirmed COVID19 person in the last 4 weeks)

**Table 3 jpm-11-00318-t003:** Case 2—Basal and after treatment laboratory parameters.

	Initial Results	After Treatment	Reference Values
Complete Blood Count			
White blood cells count	4.77	12.92	4.50–13 × 10^3^/µL
Lymphocytes	0.79	1.44	1.5–6.5 × 10^3^/µL
Neutrophils	3.64	10.85	1.8–8 × 10^3^/µL
Platelets	114,000	419,000	150.000–450.000/µL mm^3^
Hemoglobin	11.9	13.8	11.7–16.6 g/dL
Rheumathologic			
C-reactive protein	274.53	0.59	0–5 mg/L
ESR	47	6	2–15 mm/h
Procalcitonin	2.04	0.05	<0.05 ng/mL
Ferritin	3331	168	14–152 ng/mL
IL-6	113.9	1.5	<7 pg/mL
LDH	324	239	135–225 U/L
Cardiovascular			
PT	15.6	12.2	11.3–15.6 s
APPT	43.2	25	24–37 s
INR	1.18	0.91	0.84–1.2
Fibrinogen	555	182	160–390 mg/dL
D-dimers	2.38	0.27	0–0.5 µg/mL
CK	95	87	7–270 U/L
CK-MB	13.9	26.9	7–25 U/L
Troponin T	<40	<40	<40 ng/mL
NT-proBNP	6421	260	<125 pg/mL
Renal			
Creatinine	1.28	0.57	<1.2 mg/dL
BUN	76.5	41.5	<39 mg/dL
Ionogram			
Na^+^	137.6		138–145 mmol/L
K^+^	3.76		3.5–5.1 mmol/L
Hepatic			
TGO	60.3	26.9	2–48 U/L
TGP	60	145.3	2–29 U/L
GGT	107	87	3–29 U/L
Albuminemia	3.17	3.51	3.2–4.5 g/dL
Proteinemia	6.27	6.26	6–8 g/dL
Infectious			
Blood cultures	Negative		Negative
Urine culture	Negative		Negative
Influenza A + B antigen test	Negative		Negative
Quantiferon TB Gold test	Negative		Negative
PCR SARS-CoV-2	Negative		Negative
IgM SARS-CoV-2	Negative		Negative
IgG SARS-CoV-2	Positive		Negative

## Data Availability

Not applicable.

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
