# Peer review of "The Challenging Diagnosis of Pediatric Multisystem Inflammatory Syndrome Associated with Sars-Cov-2 Infection-Two Case Reports and Literature Review"

_jpm, 2021, doi:10.3390/jpm11040318_

Round 1
Reviewer 1 Report
This is a timely piece that is certain to be of interest to a select audience. It is important to share this information with a broad range of practitioners throughout the international community.
Two critical elements are necessary prior to publication.
1) The English language requires minor review and revision throughout.
2) The images included in the paper clearly identify the patients associated with the case studies. It is the recommendation of this reviewer that the patients' faces be obscured to protect their identity.
Author Response
Thank you for your review! We worked hard to send this manuscript, and we are happy to receive your answer. Definitely, we want to properly correct the requests of all reviewers and be published in your journal.
Kind regards,
Carmen Pavelescu
Reviewer 2 Report
Major comments:
- Please, write this case report in alignment with the CARE Case Report guidelines.
- Please, significantly ameliorate the way of presenting this case report using timeline tables, etc.
- Please, explain the clinical novelty of this case study: why is it interesting for the general audience; what is its importance for a journal focusing on personalized medicine (e.g., could these cases be interesting for genetic analyses?); what is its clinical impact?
Minor comments:
- “of MIS-C is lowand it has”: Please, separate “low” & “and”.
- “present illness doesn’t include”: Please, convert to “does not …”
- “Of the imagistic explorations, the chest …”: Please, convert to “imaging analysis”.
- “scan are imagistic signs demonstrating”: Please, convert to imaging findings.
- Please, improve the resolution of Figure 9.
- “Until this moment, it doesn’t exist”: Please, convert to “does not exist”.
- “even though he didn’t had”: Please, convert to “he did not have”.
- “laboratory didn’t allow us to dose the levels of IgA”: Please, convert to “did not allow us”.
Author Response
Response to Reviewer 2 Comments
Major comments:
Point 1: Please, write this case report in alignment with the CARE Case Report guidelines.
Response 1: As requested, we rewrote the reported cases according to the indications and structure from the care-statement.org website.
Point 2: Please, significantly ameliorate the way of presenting this case report using timeline tables, etc.
Response 2: We added tables and improved report boxes to make it easier to browse the material.
Point 3: Please, explain the clinical novelty of this case study: why is it interesting for the general audience; what is its importance for a journal focusing on personalized medicine (e.g., could these cases be interesting for genetic analyses?); what is its clinical impact?
Response 3: Out of a desire to improve our report, we have added the CDC Guide and reformulated the discussions and conclusions to convey why the article is interesting for scientific and medical professionals and why we hope it could be published in JPM.
Minor comments:
- “of MIS-C is lowand it has”: Please, separate “low” & “and”.
- “present illness doesn’t include”: Please, convert to “does not …”
- “Of the imagistic explorations, the chest …”: Please, convert to “imaging analysis”.
- “scan are imagistic signs demonstrating”: Please, convert to imaging findings.
- Please, improve the resolution of Figure 9.
- “Until this moment, it doesn’t exist”: Please, convert to “does not exist”.
- “even though he didn’t had”: Please, convert to “he did not have”.
- “laboratory didn’t allow us to dose the levels of IgA”: Please, convert to “did not allow us”.
Response minor comments: we followed all points from 1-8, and modified it.
Hope we have met the requirements.
Thank you in advance,
Carmen Pavelescu&Authors
Round 2
Reviewer 2 Report
.